# Improved Contrastive Divergence Training of Energy Based Models

## Abstract

We propose several different techniques to improve contrastive divergence training of energy-based models (EBMs). We first show that a gradient term neglected in the popular contrastive divergence formulation is both tractable to estimate and is important to avoid training instabilities in previous models. We further highlight how data augmentation, multi-scale processing, and reservoir sampling can be used to improve model robustness and generation quality. Thirdly, we empirically evaluate stability of model architectures and show improved performance on a host of benchmarks and use cases, such as image generation, OOD detection, and compositional generation.

## 1 Introduction

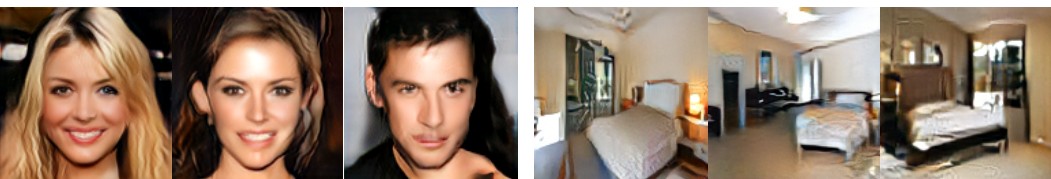

Figure 1: (Left) EBM generated 128x128 unconditional CelebA-HQ images. (Right) 128x128 unconditional LSUN Bedroom Images.

Energy-Based models (EBMs) have received an influx of interest recently and have been applied to realistic image generation (Han et al., 2019; Du & Mordatch, 2019), 3D shapes synthesis (Xie et al., 2018b) , out of distribution and adversarial robustness (Lee et al., 2018; Du & Mordatch, 2019; Grathwohl et al., 2019), compositional generation (Hinton, 1999; Du et al., 2020a), memory modeling (Bartunov et al., 2019), text generation (Deng et al., 2020), video generation (Xie et al., 2017), reinforcement learning (Haarnoja et al., 2017; Du et al., 2019), protein design and folding (Ingraham et al.; Du et al., 2020b) and biologically-plausible training (Scellier & Bengio, 2017). Contrastive divergence is a popular and elegant procedure for training EBMs proposed by (Hinton, 2002) which lowers the energy of the training data and raises the energy of the sampled confabulations generated by the model. The model confabulations are generated via an MCMC process (commonly Gibbs sampling or Langevin dynamics), leveraging the extensive body of research on sampling and stochastic optimization. The appeal of contrastive divergence is its simplicity and extensibility. It does not require training additional auxiliary networks (Kim & Bengio, 2016; Dai et al., 2019) (which introduce additional tuning and balancing demands), and can be used to compose models zero-shot.

Despite these advantages, training EBMs with contrastive divergence has been challenging due to training instabilities. Ensuring training stability required either combinations of spectral normalization and Langevin dynamics gradient clipping (Du & Mordatch, 2019), parameter tuning (Grathwohl et al., 2019), early stopping of MCMC chains (Nijkamp et al., 2019b), or avoiding the use of modern deep learning components, such as self-attention or layer normalization (Du & Mordatch, 2019). These requirements limit modeling power, prevent the compatibility with modern deep learning architectures, and prevent long-running training procedures required for scaling to larger datasets. With this work, we aim to maintain the simplicity and advantages of contrastive divergence training, while resolving stability issues and incorporating complementary deep learning advances.

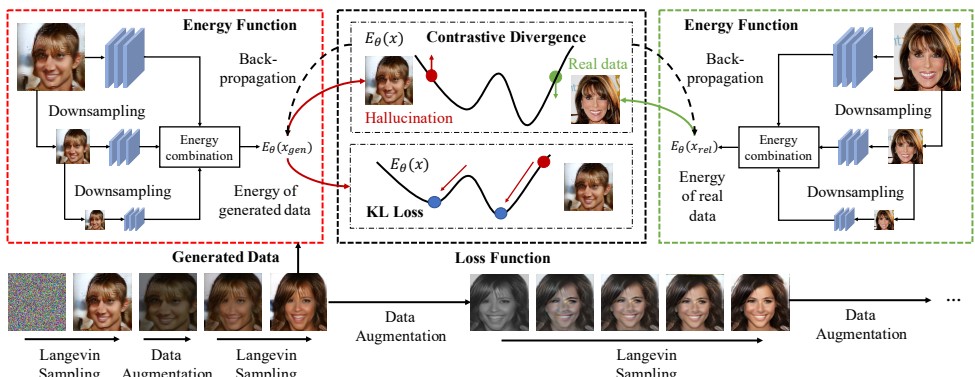

Figure 2: Illustration of our overall proposed framework for training EBMs. EBMs are trained with contrastive divergence, where the energy function decreases energy of real data samples (green dot) and increases the energy of hallucinations (red dot). EBMs are further trained with a KL loss which encourages generated hallucinations (shown as a solid red ball) to have low underlying energy and high diversity (shown as blue balls). Red/green arrows indicate forward computation for while dashed arrows indicate gradient backpropagation.

An often overlooked detail of contrastive divergence formulation is that changes to the energy function change the MCMC samples, which introduces an additional gradient term in the objective function (see Section 2.1 for details). This term was claimed to be empirically negligible in the original formulation and is typically ignored (Hinton, 2002; Liu & Wang, 2017) or estimated via high-variance likelihood ratio approaches (Ruiz & Titsias, 2019). We show that this term can be efficiently estimated for continuous data via a combination of auto-differentiation and nearest-neighbor entropy estimators. We also empirically show that this term contributes significantly to the overall training gradient and has the effect of stabilizing training. It enables inclusion of self-attention blocks into network architectures, removes the need for capacity-limiting spectral normalization, and allows us to train the networks for longer periods. We do not introduce any new objectives or complexity - our procedure is simply a more complete form of the original formulation.

We further present techniques to improve mixing and mode exploration of MCMC transitions in contrastive divergence. We propose data augmentation as a useful tool to encourage mixing in MCMC by directly perturbing input images to related images. By incorporating data augmentation as semantically meaningful perturbations, we are able to greatly improve mixing and diversity of MCMC chains. We further propose to maintain a reservoir sample of past samples, improving the diversity of MCMC chain initialization in contrastive divergence. We also leverage compositionality of EBMs to evaluate an image sample at multiple image resolutions when computing energies. Such evaluation and coarse and fine scales leads to samples with greater spatial coherence, but leaves MCMC generation process unchanged. We note that such hierarchy does not require specialized mechanisms such as progressive refinement (Karras et al., 2017)

Our contributions are as follows: firstly, we show that a gradient term neglected in the popular contrastive divergence formulation is both tractable to estimate and is important in avoiding training instabilities that previously limited applicability and scalability of energy-based models. Secondly, we highlight how data augmentation and multi-scale processing can be used to improve model robustness and generation quality. Thirdly, we empirically evaluate stability of model architectures and show improved performance on a host of benchmarks and use cases, such as image generation, OOD detection, and compositional generation.

## 2 AN IMPROVED CONTRASTIVE DIVERGENCE FRAMEWORK FOR ENERGY BASED MODELS

Energy based models (EBMs) represent the likelihood of a probability distribution for $\boldsymbol{x} \in \mathcal{R}^D$ as $p_\theta(\boldsymbol{x}) = \frac{\exp(-E_\theta(\boldsymbol{x}))}{Z(\theta)}$ where the function $E_\theta(\boldsymbol{x}) : \mathbb{R}^D \to \mathbb{R}$, is known as the *energy function*, and $Z(\theta) = \int_{\boldsymbol{x}} \exp{-E_\theta(\boldsymbol{x})}$ is known as the partition function. Thus an EBM can be represented by an neural network that takes $\boldsymbol{x}$ as input and outputs a scalar.

Training an EBM through maximum likelihood (ML) is not straightforward, as $Z(\theta)$ cannot be reliably computed, since this involves integration over the entire input domain of $\boldsymbol{x}$. However, the gradient of log-likelihood with respect to a data sample $\boldsymbol{x}$ can be represented as

$$\frac{\partial \log p_\theta(\boldsymbol{x})}{\partial \theta} = -\left( \frac{\partial E_\theta(\boldsymbol{x})}{\partial \theta} - \mathbb{E}_{p_\theta(\boldsymbol{x}')}\left[ \frac{\partial E_\theta(\boldsymbol{x}')}{\partial \theta} \right] \right). \tag{1}$$

Note that Equation 1 is still not tractable, as it requires using Markov Chain Monte Carlo (MCMC) to draw samples from the model distribution $p_\theta(\boldsymbol{x})$, which often takes exponentially long to mix. As a practical approximation to the above objective, (Hinton, 2002) proposes the contrastive divergence objective

$$\mathrm{KL}(p(\boldsymbol{x}) \,||\, p_\theta(\boldsymbol{x})) - \mathrm{KL}(\Pi_\theta^t(p(\boldsymbol{x})) \,||\, p_\theta(\boldsymbol{x})), \tag{2}$$

where $\Pi_\theta$ represents a MCMC transition kernel for $p_\theta$, and $\Pi_\theta^t(p(\boldsymbol{x}))$ represents $t$ sequential MCMC transitions starting from $p(\boldsymbol{x})$. The above objective can be seen as an improvement operator, where $\mathrm{KL}(p(\boldsymbol{x}) \,||\, p_\theta(\boldsymbol{x})) \geq \mathrm{KL}(\Pi_\theta^t(p(\boldsymbol{x})) \,||\, p_\theta(\boldsymbol{x}))$, because $\Pi_\theta$ is converging to equilibrium distribution $p_\theta(\boldsymbol{x})$ (Lyu, 2011). Furthermore, the above objective is only zero (at its fixed point), when $\Pi_\theta$ does not change the distribution of $p(\boldsymbol{x})$, which corresponds to $p_\theta(\boldsymbol{x}) = p(\boldsymbol{x})$.

## 2.1 A Missing Term in Contrastive Divergence

When taking the negative gradient of the contrastive divergence objective (Equation 2), we obtain the expression

$$-\left( \mathbb{E}_{p(\boldsymbol{x})}\left[ \frac{\partial E_\theta(\boldsymbol{x})}{\partial \theta} \right] - \mathbb{E}_{q_\theta(\boldsymbol{x}')}\left[ \frac{\partial E_\theta(\boldsymbol{x}')}{\partial \theta} \right] + \textcolor{red}{\frac{\partial q(\boldsymbol{x}')}{\partial \theta} \frac{\partial \mathrm{KL}(q_\theta(\boldsymbol{x}) \,||\, p_\theta(\boldsymbol{x}))}{\partial q_\theta(\boldsymbol{x})}} \right), \tag{3}$$

where for brevity, we summarize $\Pi_\theta^t(p(\boldsymbol{x})) = q_\theta(\boldsymbol{x})$. The first two terms are identical to those of Equation 1 and the third gradient term (which we refer to as the KL divergence term) corresponds to minimizing the divergence between $q_\theta(\boldsymbol{x})$ and $p_\theta(\boldsymbol{x})$. In practice, past contrastive divergence approaches have ignored the third gradient term, which was difficult to estimate and claimed to be empirically negligible (Hinton, 1999). These gradients correspond to a joint loss expression $\mathrm{L_{Full}}$, consisting of traditional contrastive loss $\mathrm{L_{CD}}$ and a new loss expression $\mathrm{L_{KL}}$. Specifically, we have $\mathrm{L_{Full}} = \mathrm{L_{CD}} + \mathrm{L_{KL}}$ where $\mathrm{L_{CD}}$ is

$$\mathrm{L_{CD}} = \mathbb{E}_{p(\boldsymbol{x})}[E_\theta(\boldsymbol{x})] - \mathbb{E}_{\mathrm{stop\_gradient}(q_\theta(\boldsymbol{x}'))}[E_\theta(\boldsymbol{x}')], \tag{4}$$

and the ignored KL divergence term corresponding to the loss

$$\mathrm{L_{KL}} = \mathbb{E}_{q_\theta(\boldsymbol{x})}[E_{\mathrm{stop\_gradient}(\theta)}(\boldsymbol{x})] + \mathbb{E}_{q_\theta(\boldsymbol{x})}[\log(q_\theta(\boldsymbol{x}))]. \tag{5}$$

Despite being difficult to estimate, we show that $\mathrm{L_{KL}}$ is a useful tool for both speeding up and stabilizing training of EBMs. Figure 2 illustrates the overall effects of both losses. Equation 4 encourage the energy function to assign low energy to real samples and high energy for generated samples. However, only optimizing Equation 4 often leads to an adversarial mode where the energy function learns to simply generate an energy landscape that makes sampling difficult. The KL divergence term counteracts this effect, and encourages sampling to closely approximate the underlying distribution $p_\theta(\boldsymbol{x})$, by encouraging samples to be both low energy under the energy function as well as diverse. Next, we discuss our approach towards estimating this KL divergence, and show that it significantly improves the stability when training EBMs.

## 2.2 Estimating the missing gradient term

Estimating $\mathrm{L_{KL}}$ can further be decomposed into two separate objectives, minimizing the energy of samples from $q_\theta(\boldsymbol{x})$, which we refer to as $\mathrm{L_{opt}}$ (Equation 6) and maximizing the entropy of samples from $q_\theta(\boldsymbol{x})$ which we refer to as $\mathrm{L_{ent}}$ (Equation 7).

**Minimizing Sampler Energy.** To minimize the energy of samples from $q_\theta(\boldsymbol{x})$ we can directly differentiate through both the energy function and MCMC sampling. We follow recent work in EBMs and utilize Langevin dynamics (Du & Mordatch, 2019; Nijkamp et al., 2019b; Grathwohl et al., 2019) for our MCMC transition kernel, and note that each step of Langevin sampling is fully differentiable with respect to underlying energy function parameters. Precisely, gradient of $\mathrm{L_{opt}}$ becomes

$$\frac{\partial \mathrm{L_{opt}}}{\partial \theta} = \mathbb{E}_{q_\theta(\boldsymbol{x}'_0, \boldsymbol{x}'_1, \ldots, \boldsymbol{x}'_t)}\left[ \frac{\partial E_{\mathrm{stop\_gradient}(\theta)}(\boldsymbol{x}'_{t-1} - \nabla_{\boldsymbol{x}'_{t-1}} E_\theta(\boldsymbol{x}'_{t-1}) + \omega)}{\partial \theta} \right], \quad \omega \sim \mathcal{N}(0, \lambda) \tag{6}$$

where $\boldsymbol{x}_i'$ represents the i$^{\text{th}}$ step of Langevin sampling. To reduce to memory overhead of this differentiation procedure, we only differentiate through the last step of Langevin sampling (though we show it the appendix that leads to the same effect as differentiation through Langevin sampling).

**Entropy Estimation.** To maximize the entropy of samples from $q_\theta(\boldsymbol{x})$, we use a non-parametric nearest neighbor entropy estimator (Beirlant et al., 1997), which is shown to be mean square consistent (Kozachenko & Leonenko, 1987) with root-n convergence rate (Tsybakov & Van der Meulen, 1996). The entropy $H$ of a distribution $p(\boldsymbol{x})$ can be estimated through a set $X = x_1, x_2, \ldots, x_n$ of $n$ different points sampled from $p(\boldsymbol{x})$ as $H(p_\theta(\boldsymbol{x})) = \frac{1}{n} \sum_{i=1}^{n} \ln(n \cdot \text{NN}(x_i, X)) + O(1)$ where the function $\text{NN}(x_i, X)$ denotes the nearest neighbor distance of $x_i$ to any other data point in $X$. Based off the above entropy estimator, we write L$_{\text{ent}}$ as

$$\text{L}_{\text{ent}} = \mathbb{E}_{q(\boldsymbol{x})}[\log(\text{NN}(\boldsymbol{x}, B))] \qquad (7)$$

where we measure the nearest neighbor with respect to a set $B$ of 1000 past samples from MCMC chains (see Section 2.5 for more details). We utilize L2 distance as the metric for computing nearest neighbors. Alternatively, Stein's identity may also be used to estimate entropy, but this requires considering all samples, as opposed to the nearest, becoming computationally intractable. Our entropy estimator serves a simple, quick to compute estimator of entropy, that prevents sampling from collapsing. Empirically, we find that the combination of the above terms in L$_{\text{KL}}$ significantly improves both the stability and generation quality of EBMs, improving robustness across different model architectures.

## 2.3 Data Augmentation Transitions

Langevin sampling, our MCMC transition kernel, is prone to falling into local probability modes (Neal, 2011). In the image domain, this manifests with sampling chains always converging to a fixed image (Du & Mordatch, 2019). A core difficulty is that distances between two qualitatively similar images can be significantly far away from each in input domain, on which sampling is applied. While L$_{\text{KL}}$ encourages different sampling chains to cover the model distribution, Langevin dynamics alone is not enough to encourage large jumps in finite number of steps. It is further beneficial to have an individual sampling chain have to ability mix between probability modes.

To encourage greater exploration between similar inputs in our model, we propose to augment chains of MCMC sampling with periodic data augmentation transitions that encourages movement between "similar" inputs. In particular, we utilize a combination of color, horizontal flip, rescaling, and Gaussian blur augmentations. Such combinations of augmentation has recently seen success applied in unsupervised learning (Chen et al., 2020). Specifically, during training time, we initialize MCMC sampling from a data augmentation applied to an input sampled from the buffer of past samples. At test time, during generation, we apply a random augmentation to the input after every 20 steps of Langevin sampling. We illustrate this process in the bottom of Figure 2. Data augmentation transitions are always taken.

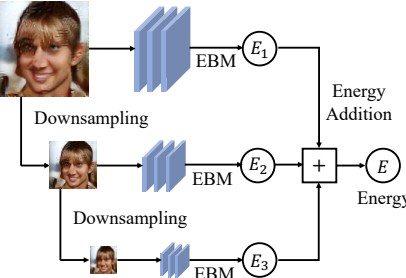

Figure 3: Illustration of our multi-scale EBM architecture. Our energy function over an image is defined compositionally as the sum of energy functions on different resolutions of an image.

## 2.4 Compositional Multi-scale Generation

To encourage energy functions to focus on features in both low and high resolutions, we define our energy function as the composition (sum) of a set of energy functions operating on different scales of an image, illustrated in Figure 3. Since the downsampling operation is fully differentiable, Langevin based sampling can be directly applied to the energy function. In our experiments, we utilize full, half, and quarter resolution image as input and show in the appendix that this improves the generation performance.

## 2.5 Reservoir Sampling

To encourage $q_\theta(\boldsymbol{x})$ to match $p_\theta(\boldsymbol{x})$, MCMC steps in $q_\theta(\boldsymbol{x})$ are often initialized from past samples from $q_\theta(\boldsymbol{x})$ to enable more diverse mode exploration, a training objective known as persistent contrastive divergence (Tieleman, 2008). Du & Mordatch (2019) propose to implement sampling

Table 1: Table of Inception and FID scores for generations of CIFAR-10, CelebA-HQ and LSUN bedroom scenes. * denotes our reimplementation of a SNGAN 128x128 model using the torch mimicry GAN library. All others numbers are taken directly from corresponding papers.

| Model | Inception* | FID |
|---|---|---|
| **CIFAR-10 Unconditional** | | |
| PixelCNN (Van Oord et al., 2016) | 4.60 | 65.93 |
| IGEBM (Du & Mordatch, 2019) | 6.02 | 40.58 |
| DCGAN (Radford et al., 2016) | 6.40 | 37.11 |
| WGAN + GP (Gulrajani et al., 2017) | 6.50 | 36.4 |
| Ours | 7.58 | 35.4 |
| SNGAN (Miyato et al., 2018) | 8.22 | 21.7 |
| **CelebA-HQ 128x128 Unconditional** | | |
| SNGAN* | - | 55.25 |
| Ours | - | 35.06 |
| SSGAN (Chen et al., 2019) | - | 24.36 |
| **LSUN Bedroom 128x128 Unconditional** | | |
| SNGAN* | - | 125.53 |
| Ours | - | 49.30 |

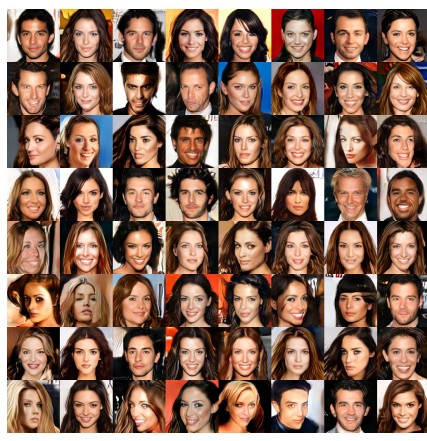

Figure 4: Randomly selected unconditional CelebA-HQ samples from our trained EBM

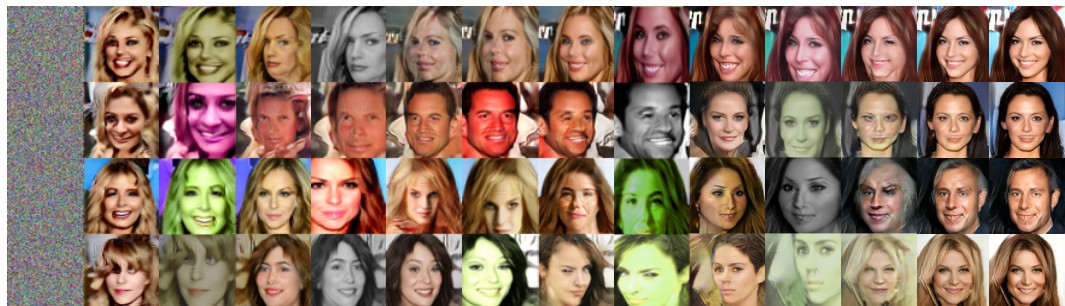

Figure 5: Visualization of Langevin dynamics sampling chains on an EBM trained on CelebA-HQ 128x128. Samples travel between different modes of images. Each consecutive images represents 30 steps of sampling, with data augmentation transitions every 60 steps .

from past samples by utilizing a replay buffer of samples from $q_\theta(x)$ interspersed with samples initialized from random noise. By storing a large batch of past samples, the replay buffer is able to enforce diversity across chains. However, as samples are initialized from the replay buffer and added to the buffer again, the replay buffer becomes filled with a set of correlated samples from $q_\theta(x)$ over time. To encourage a buffer distribution representative of all past samples, we instead use reservoir sampling technique over all past samples from $q_\theta(x)$. This technique has previously been found helpful in balancing replay in reinforcement learning (Young et al., 2018; Isele & Cosgun, 2018; Rolnick et al., 2019). Under a reservoir sampling implementation, any sample from $q_\theta(x)$ has an equal probability of being the reservoir buffer.

## 3 EXPERIMENTS

We perform empirical experiments to validate the following set of questions: (1) What are the effects of each proposed component towards training EBMs? (2) Are our trained EBMs able to perform well on downstream applications of EBMs (generation, compositionality, out-of-distribution detection)? We provide ablations of each of our proposed components in the appendix.

### 3.1 EXPERIMENTAL SETUP

We investigate the efficacy of our proposed approach. Models are trained using the Adam Optimizer (Kingma & Ba, 2015), on a single 32GB Volta GPU for CIFAR-10 for 1 day, and for 3 days on 8 32GB Volta GPUs for CelebaHQ, and LSUN datasets. We provide detailed training configuration details in the appendix.

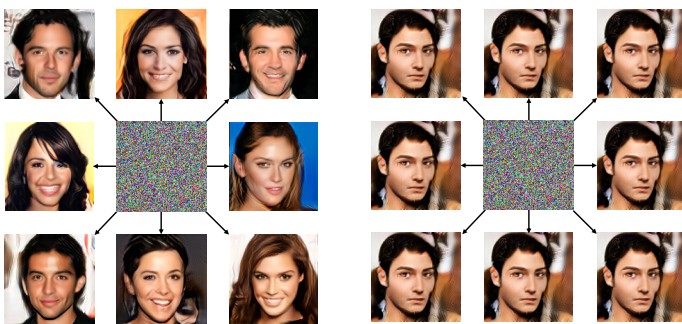

Figure 6: Output samples of running Langevin Dynamics from a fixed initial sample (center of square), with or without intermittent data-augmentation transitions. Without data-augmentation transitions, all samples converge to same image, while data augmentations enables chains to seperate

Our improvements are largely built on top of the EBMs training framework proposed in (Du & Mordatch, 2019). We use a buffer size of 10000, with a resampling rate of 5% with L2 regularization on output energies. Our approach is significantly more stable than IGEBM, allowing us to remove aspects of regularization in (Du & Mordatch, 2019). We remove the clipping of gradients in Langevin sampling as well as spectral normalization on the weights of the network. In addition, we add self-attention blocks and layer normalization blocks in residual networks of our trained models. In multi-scale architectures, we utilize 3 different resolutions of an image, the original image resolution, half the image resolution and a quarter the image resolution. We report detailed architectures in the appendix. When evaluating models, we utilize the EMA model with EMA weight of 0.999.

## 3.2 IMAGE GENERATION

We evaluate our approach on CIFAR-10, LSUN bedroom (Yu et al., 2015), and CelebA-HQ (Karras et al., 2017) datasets and analyze our characteristics of our proposed framework. Additional analysis and ablations can be found in the appendix of the paper.

**Image Quality.** We evaluate our approach on unconditional generation in Table 1. On CIFAR-10 we find that approach, while not being state-of-the-art, significantly outperforms past EBM approaches that based off implicit sampling from an energy landscape (with approximately the same number of parameters), and has performance in the range of recent GAN models on high resolution images. We further present example qualitative images from CelebA-HQ in Figure 11b and present qualitative images on other datasets in the appendix of the paper. We note that our reported SNGAN performance on CelebA-HQ and LSUN Bedroom use default hyperparameters from ImageNet models. Gaps in performance with our model are likely smaller with better dataset specific hyper-parameters.

**Effect of Data Augmentation.** We evaluate the effect of data augmentation on sampling in EBMs. In Figure 5 we show that by combining Langevin sampling with data augmentation transitions, we are able to enable chains to mix across different images, whereas prior works have shown Langevin converging to fixed images. In Figure 6 we further show that given a fix random noise initialization, data augmentation transitions enable to reach a diverse number of different samples, while sampling without data augmentation transitions leads all chains to converge to the same face.

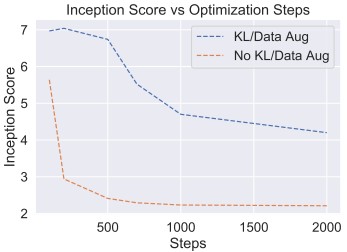

Figure 7: Illustration of Inception Score over long run chains with or without data augmentation/KL loss.

**Mode Convergence.** We further investigate high likelihood modes of our model. In Figure 8, we compare very low energy samples (obtained after running gradient descent 1000 steps on an energy function) for both our model with data augmentation and KL loss and a model without either term. Due to improved mode exploration, we find that low temperature samples under our model with data augmentation/KL loss reflect typical high likelihood "modes" in the training dataset, while our baseline models converges to odd shapes, also noted in (Nijkamp et al., 2019a). In Figure 7, we quantitatively measure Inception scores as we run steps of gradient descent with or without data augmentation and KL loss. Our

Low Temperature Samples (Data Aug + KL)

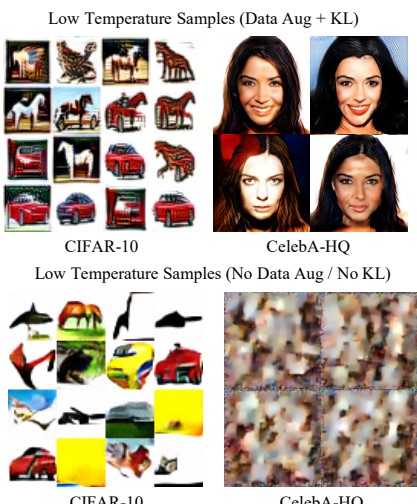

CIFAR-10                    CelebA-HQ

Low Temperature Samples (No Data Aug / No KL)

CIFAR-10                    CelebA-HQ

Figure 8: Illustration of very low temperature samples from our model with KL loss and data augmentation (top) on CIFAR-10 and CelebA-HQ compared to without (bottom). After a large number of sampling steps, models trained without KL/data augmentation converge to stranges hues in CIFAR-10 and random textures on CelebA-HQ. In contrast, due to better mode exploration, adding both losses maintain naturalistic image modes on both CIFAR-10 and CelebA-HQ.

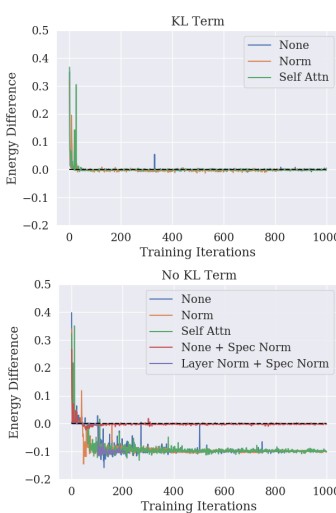

Figure 9: The KL loss significantly improves the stability of EBM training. Stable EBM training occurs when the energy difference is roughly zero (indicated by dashed black line). We find that without using the KL loss term EBM training diverges quickly with different additions to network architecture, while with the KL loss training is stable. Spectral normalization improves stability of training, but addition of components such as layer normalization also destabilizes training.

Inception score decreases much more slowly, with some degree of degradation expected since low temperature samples have less diversity.

**Stability/KL Loss** EBMs are known to difficult to train, and to be sensitive to both the exact architecture and to various hyper-parameters. We found that the addition of a KL term into our training objective significantly improved the stability of EBM training, by encouraging the sampling distribution to match the model distribution. In Figure 9, we measure the energy difference between real and generated images over the course training when adding normalization and self-attention layers to models. We find that with $L_{kl}$, the energy difference between both is kept at 0, which indicates stable training. in contrast, without $L_{kl}$ all models, with the exception of a model with spectral normalization, diverge to a negative energy different of $-1$, indicating training collapse. Furthermore, the use of spectral normalization by itself, albeit stable, precludes the addition of other modern network components such as layer normalization. The addition of the KL term itself is not too expensive, simply requiring an additional nearest neighbor computation during training, which can be relatively insignificant cost compared to the number of negative sampling steps used during training. With a intermediate number of negative sampling steps (60 steps) during training, adding the KL term roughly twice as slow as normal training. This difference is decreased with larger number of sampling steps. Please see the appendix for additional analysis of gradient magnitudes of $L_{kl}$ and $L_{cd}$.

**Ablations.** We ablate each portion of our proposed approach in Table 2. We find that the KL loss is crucial to the stability of training an EBM, and find that additions such as a multi-scale architecture are not stable without the presence of a KL loss.

Table 2: Effect of each ablation on overall image generation on CIFAR-10. We further report overall stability of training. KL Loss has a significant effect on stability of EBM training.

| KL Loss ($L_{opt}$) | KL Loss ($L_{ent}$) | Data Aug | Reservoir Sampling | Multiscale | Inception Score | FID | Stability |
|---|---|---|---|---|---|---|---|
| No | No | No | No | No | 1.46 | 253.1 | No |
| No | No | Yes | No | No | 6.19 | 61.3 | No |
| Yes | No | Yes | No | No | 6.28 | 59.8 | Yes |
| Yes | Yes | Yes | No | No | 6.53 | 54.3 | Yes |
| Yes | Yes | Yes | No | Yes | 6.78 | 50.8 | Yes |
| Yes | Yes | Yes | Yes | Yes | 7.48 | 35.4 | Yes |

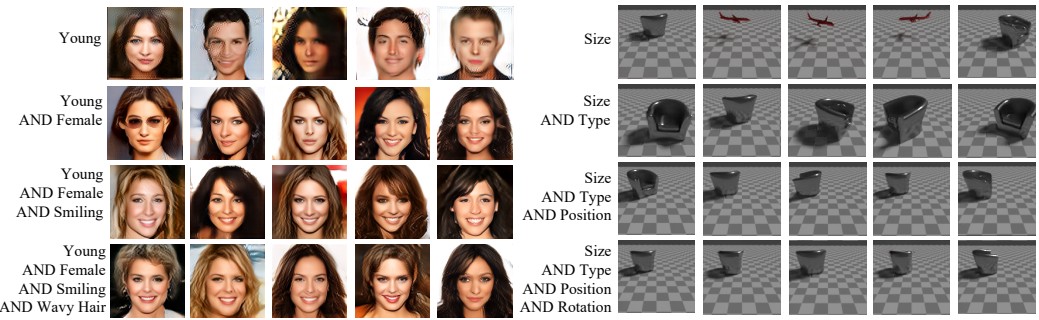

Figure 10: Examples of compositionality in face attributes (left) and different renderings of a shape (right). Our model is able to construct high resolution globally coherent compositional renderings, including fine detail such as lighting and reflections (right).

Table 3: Table of out-of-distribution detection scores on unconditional models trained on CIFAR-10 using $\log(p_\theta(x))$.. Our approach performs the best out of both likelihood and EBM models. *JEM is not directly comparable as it uses supervised labels.

| Model | PixelCNN++ | Glow | IGEBM | JEM* | Ours |
|---|---|---|---|---|---|
| SVHN | 0.32 | 0.24 | 0.63 | 0.67 | **0.91** |
| Textures | 0.33 | 0.27 | 0.48 | 0.60 | **0.88** |
| CIFAR10 Interpolation | **0.71** | 0.51 | 0.70 | 0.65 | 0.65 |
| CIFAR100 | 0.63 | 0.55 | 0.50 | 0.67 | **0.83** |
| Average | 0.50 | 0.39 | 0.57 | 0.65 | **0.82** |

## 3.3 COMPOSITIONALITY

Energy Based Models (EBMs) have the ability to *compose* with other models at generation time (Hinton, 1999; Haarnoja et al., 2017; Du et al., 2020a). We investigate to what extent EBMs trained under our new proposed framework can also exhibit compositionality. See (Du et al., 2020a) for a discussion of various compositional operators and applications in EBMs. In particular, we train *independent* EBMs $E(x|c_1)$, $E(x|c_2)$, $E(x|c_3)$, that learn conditional generative distribution of concept factors $c$ such as facial expression or object position. We test to see if we can compose independent energy functions together to generate images with each concept factor simultaneously. We consider compositions the CelebA-HQ dataset, where we train independent energy functions of face attributes of age, gender, smiling, and wavy hair and a high resolution rendered of different objects rendered at different locations, where we train an energy function on size, position, rotation, and identity of the object.

**Qualitative Results.** We present qualitative results of compositions of energy functions in Figure 10. In both composition settings, our approach is able to successfully generate images with each of conditioned factors, while also being globally coherent. The left image shows that as we condition on factors of young, female, smiling, and wavy hair, images generation begins exhibiting each required feature. The right image similarly shows that as we condition on factors of size, type, position, and rotation, image generations begin to exhibit each conditioned attribute. We note that figures are visually consistent in terms of lighting, shadows and reflections. We note that generations of thee combination of different factors are *only* specified at generation time, with models being trained *independently*. Our results indicate that our framework for training EBMs is a promising direction for high resolution compositional visual generation. We further provide visualization of best comparative compositional model from (Vedantam et al., 2018) in the appendix and find that our approach significantly outperforms it.

## 3.4 OUT OF DISTRIBUTION ROBUSTNESS

Energy Based Models (EBMs) have also been shown to exhibit robustness to both out-of-distribution and adversarial samples (Du & Mordatch, 2019; Grathwohl et al., 2019). We evaluate out-of-distribution detection of our trained energy through log-likelihood using the evaluation metrics proposed in Hendrycks & Gimpel (2016). We similarly evaluate out-of-distribution detection of an unconditional CIFAR-10 model.

**Results.** We present out-of-distribution results in Table 3, comparing with both other likelihood models and EBMs and using log-likelihood to detect outliers. We find that on the datasets we evaluate, our approach significantly outperforms other baselines, with the exception of CIFAR-10 interpolations. We note the JEM (Grathwohl et al., 2019) further requires supervised labels to train the energy function, which has to shown to improve out-of-distribution performance. We posit that by more efficiently exploring modes of the energy distribution at training time, we are able to reduce the spurious modes of the energy function and thus improve out-of-distribution performance.

## 4 RELATED WORK

Our work is related to a large, growing body of work on different approaches for training EBMs. Our approach is based on contrastive divergence (Hinton, 2002), where an energy function is trained to contrast negative samples drawn from a model distribution and from real data. In recent years, such approaches have been applied to the image domain (Xie et al., 2016; Gao et al., 2018; Du & Mordatch, 2019; Nijkamp et al., 2019b; Grathwohl et al., 2019). (Gao et al., 2018) also proposes a multi-scale approach towards generating images from EBMs, but different from our work, uses each sub-scale EBM to initialize the generation of the next EBM. Our work builds on existing works towards contrastive divergence based training of EBMs, and presents improvements in generation and stability.

A difficulty with contrastive divergence training is the difficulty of negative sample generation. To sidestep this issue, a separate line of work utilizes an auxiliary network to amortize the negative portions of the sampling procedure (Kim & Bengio, 2016; Kumar et al., 2019; Han et al., 2019; Xie et al., 2018a; Song & Ou, 2018). One line of work (Kim & Bengio, 2016; Kumar et al., 2019; Song & Ou, 2018), utilizes a separate generator network for negative image sample generations. In contrast, (Xie et al., 2018a), utilizes a generator to warm start generations for negative samples and (Han et al., 2019) minimizes a divergence triangle between three models. While such approaches enable better qualitative generation, they also lose some of the flexibility of the EBM formulation. For example, separate energy models can no longer be composed together for generation.

In addition, other approaches towards training EBMs seek instead to investigate separate objectives to train the EBM. One such approach is score matching, where the gradients of an energy function are trained to match the gradients of real data (Hyvärinen, 2005; Song & Ermon, 2019), with a related denoising (Sohl-Dickstein et al., 2015; Saremi et al., 2018; Ho et al., 2020) approach. Additional objectives include noise contrastive estimation (Gao et al., 2020) and learned Steins discrepancy (Grathwohl et al., 2020).

Most prior work in contrastive divergence has ignored the KL term (Hinton, 1999; Salakhutdinov & Hinton, 2009). A notable exception is (Ruiz & Titsias, 2019), which obtains a similar KL divergence term to ours. Ruiz & Titsias (2019) use a high variance REINFORCE estimator to estimate the gradient of the KL term, while our approach relies on auto-differentiation and nearest neighbor entropy estimators. Differentiation through model generation procedures has previously been explored in other models (Finn & Levine, 2017; Metz et al., 2016). Other related entropy estimators include those based on Stein's identity (Liu et al., 2017) and MINE (Belghazi et al., 2018). In contrast to these approaches, our entropy estimator relies only on nearest neighbor calculation, and does not require the training of an independent neural network.

## 5 CONCLUSION

We propose a simple and general framework for improving generation and ease of training with energy based models. We show that the framework enables high resolution compositional image generation and out-of-distribution robustness. In the future, we are interested in further computational scaling of our framework, and applications to domains such as text and reasoning.

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

## A APPENDIX

### A.1 MODEL ARCHITECTURES

We list model architectures used in our experiments in Figure 11. When training multi-scale energy functions, our final output energy function is the sum of energy functions applied to the full resolution image, half resolution image, and quarter resolution image. We use the architecture reported in Figure 11 for the full resolution image. The half-resolution models shares the architecture listed in Figure 11, but with all layers before and including the first down-sampled residual block removed. Similarily, the quarter resolution models share the architectures listed, but with all layers before two down-sampled residual blocks removed.

| 3x3 conv2d, 64 |
| --- |
| ResBlock 64 |
| ResBlock Down 64 |
| ResBlock 64 |
| ResBlock Down 64 |
| Self Attention 64 |
| ResBlock 128 |
| ResBlock Down 128 |
| ResBlock 256 |
| ResBlock Down 256 |
| Global Mean Pooling |
| Dense $\rightarrow$ 1 |

(a) The model architecture used for CIFAR-10 experiments.

| 3x3 conv2d, 64 |
| --- |
| ResBlock Down 64 |
| ResBlock Down 128 |
| ResBlock Down 128 |
| ResBlock 256 |
| ResBlock Down 256 |
| Self Attention 512 |
| ResBlock 512 |
| ResBlock Down 512 |
| Global mean Pooling |
| Dense $\rightarrow$ 1 |

(b) The model architecture used for CelebA/LSUN room experiments.

Figure 11: Architecture of models on different datasets.

### A.2 EXPERIMENT CONFIGURATIONS FOR DIFFERENT DATASETS

**CIFAR-10**    For CIFAR-10, we use 40 steps of Langevin sampling to generate a negative sample. The Langevin sampling step size is set to be 100, with Gaussian noise of magnitude 0.001 at each iteration. The data augmentation transform consists of color augmentation of strength 1.0 from (Chen et al., 2020), as a random horizontal crop, and a image resize between 0.3 and 1.0 and a Gaussian blur of 5 pixels.

**CelebA/LSUN Bed**    For CelebA and LSUN bed datasets, we use 40 steps of Langevin sampling to generate negative samples. The Langevin sampling step size is set to be 1000, with Gaussian noise of magnitude 0.001 applied at each iteration. The data augmentation transform consists of color augmentation of strength 0.5 from (Chen et al., 2020), as a random horizontal crop, and a image resize between 0.3 and 1.0 and a Gaussian blur of 11 pixels.

### A.3 COMPARISON OF CD/KL GRADIENT MAGNITUDES

We plot the overall gradient magnitudes of the contrastive divergence and KL loss terms during training of an EBM in Figure 12. We find that relative magnitude of both training remains constant across training, and that the gradient of the KL objective is non-negligible.

### A.4 ANALYSIS OF TRUNCATED LANGEVIN BACKPROPAGATION

To test the effect of truncating backpropogation through the KL loss to only one sampling step of Langevin sampling, we train two seperate models on MNIST, one with backpropogation through

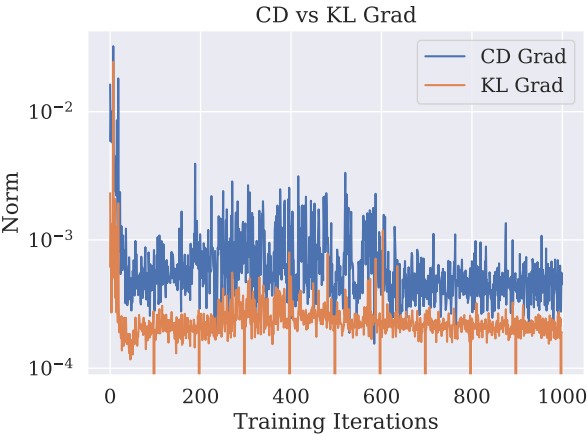

Figure 12: Plots of the gradient magnitude of $\mathcal{L}_{\text{KL}}$ and $\mathcal{L}_{\text{CD}}$ across training iterations. Influences and relative magnitude of both loss terms stays constant through training.

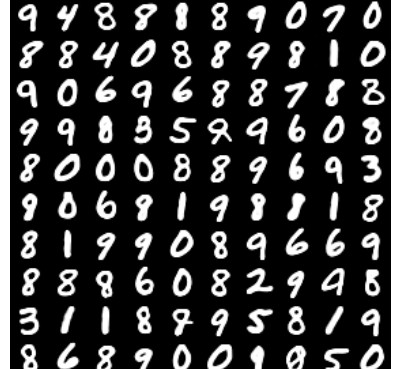

Figure 13: Generations on MNIST with backpropogation through 1 step of Langevin sampling.

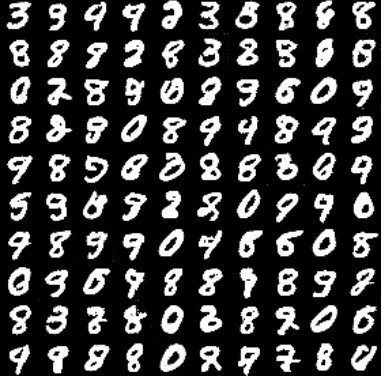

Figure 14: Generations on MNIST with backpropogation through all steps of Langevin sampling.

all Langevin steps, and one with backpropagation through only the last Langevin step. We obtain FIDs of 90.54 with backpropogation through only 1 step of Langevin sampling and FIDs of 94.85 with backpropogation through all steps of Langevin sampling. We present illustrations of samples generated with one step in Figure 13 and with all steps in Figure 14.

## A.5 ANALYSIS OF EFFECT OF KL LOSS ON MODE EXPLORATION

The KL loss adds an additional term to EBM training that encourages EBM training updates to maintain good mode coverage while optimizing the usual contrastive divergence objective. Thus the KL loss serves as a regularizer to prevent EBM sampling from collapsing. In the absence of the KL loss, EBM sampling always eventually collapses and generates samples in Figure 15

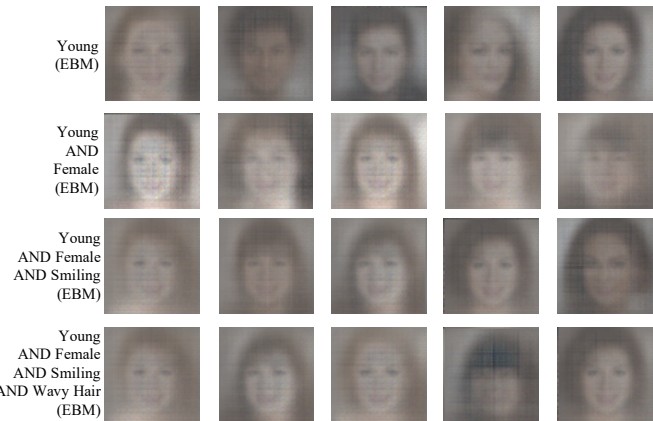

Figure 16: Illustrations of compositional generations from (Vedantam et al., 2018). Generations are significantly more blurry than our generations.

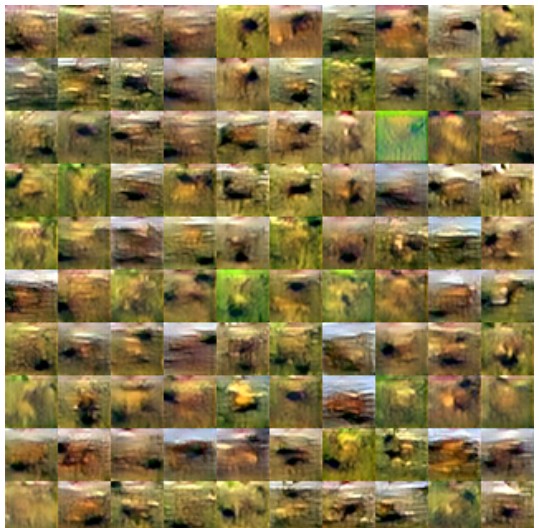

Figure 15: Illustration of collapsed sampling from an EBM.

## A.6 COMPARISON TO OTHER COMPOSITIONAL GENERATIVE MODELS

To our knowledge, there are relatively few other models that can compositionally combine, with the approach of JVAE (Vedantam et al., 2018) being the closest to our work. We provide comparisons in Figure 16. Our approach is significantly less blurry than JVAE.

## A.7 ADDITIONAL QUALITATIVE IMAGES

We present randomly generated qualitative images on the LSUN dataset in Figure 17 and the CIFAR-10 dataset in Figure 18. In both setting, we find that unconditional images appear mostly globally coherent.

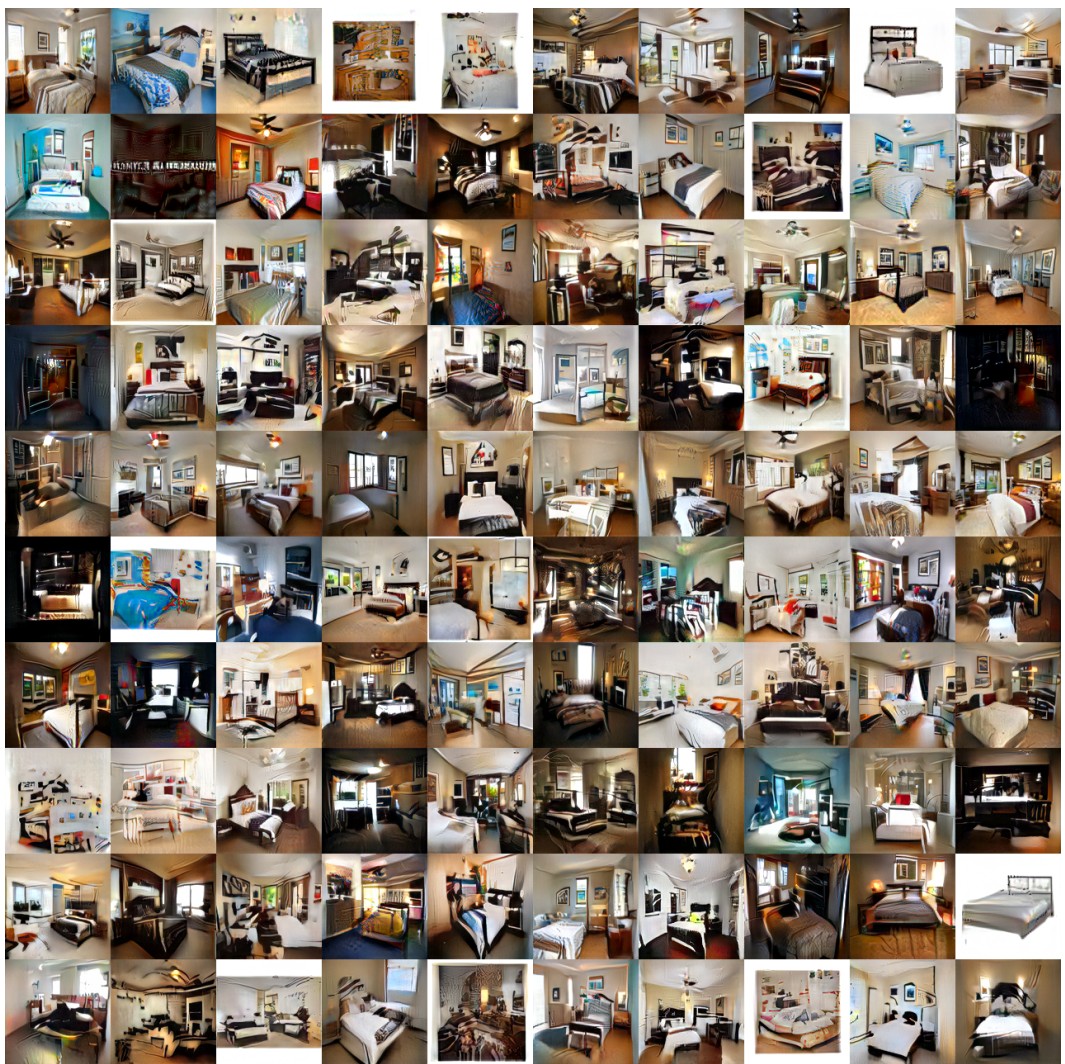

Figure 17: Randomly selected unconditional LSUN bed 128x128 samples from our trained EBM.

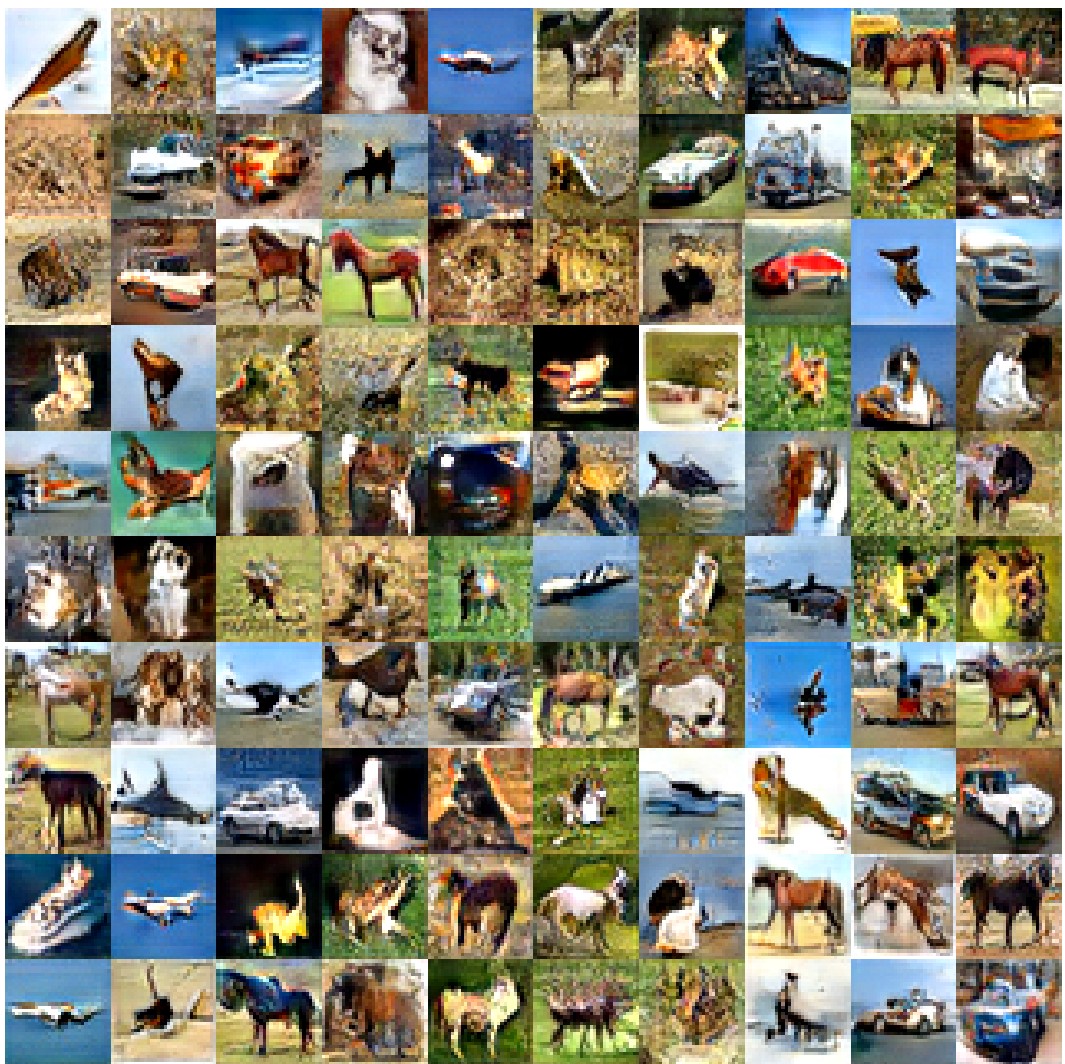

Figure 18: Randomly selected unconditional CIFAR-10 samples from our trained EBM.

