# OpenReview forum: "Improved Contrastive Divergence Training of Energy Based Models"
_ICLR.cc/2021/Conference — Reject_

### Official Review · AnonReviewer3 · 2020-10-28
**limited novelty and reference missing**

**Rating:** 4
**Confidence:** 5

**Review:**

Review: This paper studies how to improve contrastive divergence (CD) training of energy-based models (EBMs) by revisiting the gradient term neglected in the traditional CD learning. This paper also introduces some useful techniques, such as data augmentation, multi-scale energy design, and reservoir sampling to improve the training of energy-based model. Empirical studies are performed to validate the proposed learning strategy on the task of image generation, OOD detection, and compositional generation.

Strength:
+ The idea of dealing with the missing term in the traditional CD learning is important and relevant.
+ The paper is well written. Specifically, the figure illustration and the organization of the paper make me feel quite easy to follow the paper.
+ The motivation of the method is clear, and the experimentation looks OK.

Concerns:
+ The contribution of the paper is quite limited. Even though this paper tried to estimate the missing term in the CD learning, it lacks a comprehensive analysis of the benefit and advantages of doing so. For example, (1) what is the cost to add such a term? (2) Can you validate theoretically such a missing term can be helpful for MCMC mixing as you claimed in the paper?

+ About motivation. Even though the motivation of the current paper is clear, which is to improve the CD learning. However, the CD learning (in equation 2) is biased compared with MLE (in equation 1). The original motivation for CD learning is to make EBM learning more efficient. Since currently there has been EBM training method without MCMC or with amortized sampling, I am not sure if the current method is still useful for the community.

+ About synthesis quality: the synthesized images generated by the proposed images are not impressive. Artifacts can be obviously observed in Figure 12.

+ Missing important references in related works. The current paper missed to cite the pioneering paper about MLE training of ConvNet-EBM [1]. Those EBM papers you have cited from 2019 is based on [1] or its variant.

+ Incomplete narrative of the development of EBMs in the introduction.  Even though the narrative of the development of EBMs is quite comprehensive, it is not complete and even a little bit misleading. For example, since 2016, the EBMs have been applied to realistic image generation (2016-2019)[1, 3, 4, 6, 7], video generation (2017)[2, 4], and 3D generation (2018)[5] in the community of computer vision. Therefore, the current research direction seems not to be novel, given the fact that authors might miss a lot important developments about EBM made by other fields. CD learning is also studied and discussed in [1] for deep EBM. The current papers only discussed and connected EBM development happened recently in ML community (2019, 2020).

+ typo: in Section 2.1 line 16: KL divergenceterm => KL divergence term

Some references:
+ [1] A Theory of Generative ConvNet (ICML 2016)
+ [2] Synthesizing Dynamic Pattern by Spatial-Temporal Generative ConvNet (CVPR 2017)
+ [3] Cooperative Learning of Energy-Based Model and Latent Variable Model via MCMC Teaching (AAAI 2018)
+ [4] Cooperative learning of descriptor and generator networks. IEEE Transactions on Pattern Analysis and Machine Intelligence (PAMI 2018).
+ [5] Learning Descriptor Networks for 3D Shape Synthesis and Analysis. (CVPR 2018)
+ [6] Learning generative ConvNets via multigrid modeling and sampling. (CVPR 2018)
+ [7] Divergence triangle for joint training of generator model, energy-based model, and inference model. (CVPR 2019)

---

> ### Author Response · Authors · 2020-11-20
> **Response**
>
> Thank you for your comments. We agree the mentioned papers are related and we have added them  in the paper (applications in the Introduction section and more method based papers in the Related Work section)
>
> Q1) The contribution of the paper is quite limited. Even though this paper tried to estimate the missing term in the CD learning, it lacks a comprehensive analysis of the benefit and advantages of doing so. For example, (1) what is the cost to add such a term? (2) Can you validate theoretically such a missing term can be helpful for MCMC mixing as you claimed in the paper?
>
> We have added analysis to the cost of adding the KL term (training is roughly 1x slower) in section 3.2 (stability/KL loss). The overall cost comes primarily from loading the 1000 nearest neighbors for entropy calculation.
>
> We can theoretically see that such a missing term is helpful for MCMC mixing is the objective function is directly the KL divergence between the MCMC distribution and the model distribution, which by definition is minimized when MCMC mixing is the entire model distribution.
>
>
> Q2) About motivation. Even though the motivation of the current paper is clear, which is to improve the CD learning. However, the CD learning (in equation 2) is biased compared with MLE (in equation 1). The original motivation for CD learning is to make EBM learning more efficient. Since currently there has been EBM training method without MCMC or with amortized sampling, I am not sure if the current method is still useful for the community.
>
> While there are approaches towards training EBM without either MCMC or amortized sampling, MCMC based training of EBMs enables generation through MCMC. This enables considerable flexibility in generation, such as the ability to compositionally generate images from multiple models (section 3.3), of which we show the best compositional generation results that of an model that we are aware of. Furthermore, this enables EBMs to be robust and performance well at out-of-distribution detection (which we also outperform any other method we are aware of).
>
> Q3) Related Work
> We have revised both our narrative and missing references. We have changed our narrative to discuss each of the papers you have cited.
>
> Q4) About synthesis quality: the synthesized images generated by the proposed images are not impressive. Artifacts can be obviously observed in Figure 12.
>
> Our generated image quality is the best we have seen for a sampling method based only on implicit generation on the energy landscape and seems to outperform images seen in [1,2,3,4]. We note that out of the presented images (including un-curated ones in Figure 4) ,only the ones in Figure 12 have obvious artifacts.
>
> [1] Implict Generation and Generalization with Energy Based Models. NeurIPS 2019.
>
> [2] Learning generative ConvNets via multigrid modeling and sampling. (CVPR 2018)
>
> [3] Cooperative Learning of Energy-Based Model and Latent Variable Model via MCMC Teaching (AAAI 2018)
>
> [4]  Theory of Generative ConvNet (ICML 2016)

---

> ### Comment · AnonReviewer3 · 2020-11-20
> **Typos and corrupted references**
>
> Thanks for the update.   I found some references are corrupted in the introduction part, for examples: "3D shapes synthesis ()" and "video generation ()".
>
> Also, some typos in  the revised related work, e.g.,  "where a energy function" => "where an energy function". Please double check if the uploaded revised paper is the right version.

---

> > ### Author Response · Authors · 2020-11-20
> > **Thanks**
> >
> > Thanks for catching that. We unintentionally uploaded an old version of our manuscript. Typos and references should now be fixed.

---

### Official Review · AnonReviewer1 · 2020-10-29

**Rating:** 5
**Confidence:** 4

**Review:**

The paper proposes a series of new techniques to enhance the training of an energy-based model, and the proposed techniques include: adding the often neglected KL term to the training scope/data augmentation + multi-scale energy function/an experience replay buffer for training.

The experiments demonstrate the proposed method could generate high-quality images, compositional tasks and perform out-of-distribution detection.

The main idea and motivation are well and clearly conveyed by the writing.

The paper would be stronger if the authors could provide the following pieces:
- how well is the entropy estimation? We all know that estimating the entropy of data distribution from a high-dimenstional space is very difficult. Does this form of nearest neighbor applicable in other areas? It would also be great if the authors could provide some theoretical analysis here.
- while the main contribution of the paper seems to be the KL term added into the objective, there are a few other tenichques tagging along. It is not clear what role each of these techniques plays in the experiments. I recommend the authors to show an ablation study.
- Figure 9 would need to compare against other methods. It is not clear to me how the arithmetic results are stronger than the other published results.
- An important argument in the paper is that the added KL term enhances the mode coverage. Could the authors provide some more evidence on this point?

---

> ### Author Response · Authors · 2020-11-20
> **Response**
>
> Q1) how well is the entropy estimation? We all know that estimating the entropy of data distribution from a high-dimenstional space is very difficult. Does this form of nearest neighbor applicable in other areas? It would also be great if the authors could provide some theoretical analysis here.
>
> First, please note that the benefit of our nearest neighbor entropy estimator is not so much in helping generate diverse chains (Langevin dynamics with data augmentation aims to do that), but in calculating L_KL term and preventing training destabilization of energy network if sample chains collapse (i.e. there is still a non-zero gradient on energy network weights pushing the samples away from the buffer of what was previously generated if samples collapse). That said, our entropy estimation term can be directly used in other data distributions in high dimensional space, by doing a similar nearest neighbor query in other high-dimensional spaces. Analysis for the nearest neighbor entropy estimator is given at [1], which we have referenced. To reiterate:
>
> 1) Our entropy estimator exhibits a sqrt(n) convergence (where n is the number of nearest neighbor) when modeling distributions with exponential tails.
> 2) The estimator is mean square consistent to the empirical entropy estimate
>
> It is difficult to compare with recent approaches towards estimating entropy based on learned networks as there is limited theoretical analysis on their convergence properties. However, our estimator is unbiased compared to the neural network estimators but is likely less sample efficient (since neural networks can represent points of high likelihood for data) .
>
> [1] Jan Beirlant, E.J. Dudewicz, L. Gyor, E.C. van der Meulen Nonparametric Entropy Estimation: An Overview
>
>
>
> Q2) while the main contribution of the paper seems to be the KL term added into the objective, there are a few other techniques tagging along. It is not clear what role each of these techniques plays in the experiments. I recommend the authors to show an ablation study.
>
>
>
> We had ablation experiments in the appendix in our original submission and we have now moved them from the appendix to the main paper (Table 2). Overall, we find that the KL term improves the overall generation of samples. Furthermore, we found that only with the addition of KL term could we reliably train a multiscale model as train models with data augmentation for a long time.
>
> Q3) Figure 9 would need to compare against other methods. It is not clear to me how the arithmetic results are stronger than the other published results.
>
> We note that past works in addition with generative models have been limited to low resolution images[1], while we are the first work to show high resolution compositionality using the generative models. We provide comparisons with [1] in Figure 16. Our generated results are significantly higher resolution. We believe that the high resolution compositional generation is one of the most promising applications of our approach and note that [3] also quantitatively shows that energy based models significantly outperforms [1].
>
>
> Q4) An important argument in the paper is that the added KL term enhances the mode coverage. Could the authors provide some more evidence on this point?
>
> The KL term serves as a regularizer to prevent model weights from entering an area in weight space where sampling has poor mode coverage -- not necessarily that mode coverage is better compared to another healthily trained model.  Without the KL term, after a long period of training, sampling always collapses in EBMs [2], which we illustrate in Appendix A.5. The KL term prevents such a possible sampling collapse.
>
>
> [1] Ramakrishna Vedantam, Ian Fischer, Jonathan Huang, Kevin Murphy Generative Models of Visually Grounded Imagination. ICLR 2018
> [2] Grawthawl et. al. Your Classifier is Secretly an Energy Based Model. ICLR 2020
> [3] Yilun Du, Shuang Li, Igor Mordatch. Compositional Visual Generation with Energy Based Models

---

### Official Review · AnonReviewer4 · 2020-10-30
**Improved version of CD**

**Rating:** 5
**Confidence:** 5

**Review:**

This paper proposed an improved version of contrastive divergence learning of energy-based models by combining a bag of techniques: (1) add back a KL term that is neglected by previous methods (2) data augmentation (3) multi-scale processing (4) reservoir sampling. Experiments demonstrate the effectiveness of the improvements.

Pro:
The paper is well-written and easy to follow. Various experiments are performed to demonstrate the efficacy of the improved method.

Cons:
1. The advantage of adding the KL term is not quite obvious given the current experiments. The only experiment that isolates the effect of the KL term is Figure 8 (stability of training), which can be accomplished by simply adding spectral normalization. For all the other improvements, I tend to believe they are due to the techniques of (2)(3)(4).

2. For the first term of L_KL, it is not entirely correct to take gradient only over the last step of Langevin sampling. Need more justifications. For the second term, it requires computing on 1000 samples per update, where the efficiency should be discussed.

3. The multi-scale processing of EBMs has been explored in [1], which should be discussed and compared. Besides, [2][3][4] are relevant references of training EBMs that should be discussed.

4. Qualitative speaking, long-run chains in figure 7 still have a trend of degradation from realistic images. Quantitative analysis (e.g., German-Rubin statistics) would be helpful for evaluating the long-run chains clearly.

Overall, the paper proposes effective improvements on contrastive divergence of EBMs, and performs various experiments to demonstrate the efficacy. However, I am concerned about the correctness and the necessity of adding the gradient term (L_KL), which is one of the major contributions that the authors claim. Please address my concern as listed above.

[1] Learning Energy-Based Models as Generative ConvNets via Multi-grid Modeling and Sampling, Gao et al.
[2] A Theory of Generative ConvNet, Xie et al.
[3] Flow Contrastive Estimation of Energy-Based Models, Gao et al.
[4] Learning the Stein Discrepancy for Training and Evaluating Energy-Based Models without Sampling, Grathwohl et al.

---

> ### Author Response · Authors · 2020-11-20
> **Response**
>
> We thank the reviewer for their thorough and insightful review. We agree the mentioned papers are related to our work and we now discuss the mentioned papers in the related work of the rebuttal version.
>
>
> Q1) The advantage of adding the KL term is not quite obvious given the current experiments. The only experiment that isolates the effect of the KL term is Figure 8 (stability of training), which can be accomplished by simply adding spectral normalization. For all the other improvements, I tend to believe they are due to the techniques of (2)(3)(4).
>
> We had ablation experiments in the appendix in our original submission and we have now moved them to the main paper (Table 2). We find that the KL loss separately improves generation performance. Furthermore, both techniques (2) and (3) can only be used in conjunction with the KL loss -- otherwise the training process is unstable and neither techniques (2) or (3) contribute to the final results. All results in Figure 5 and 7 are based  on the presence of the KL loss. The addition of spectral normalization does not enable techniques (2) or (3).
>
> Q2) For the first term of L_KL, it is not entirely correct to take gradient only over the last step of Langevin sampling. Need more justifications. For the second term, it requires computing on 1000 samples per update, where the efficiency should be discussed.
>
> We have added comparisons to test the efficiency of computing the nearest 1000 samples in the revised paper in section 3.2 (stability/KL loss). The overall computational cost of adding the KL loss is roughly the same as generating a negative sample using 60 steps of Langevin (slowing training down by a factor of 1). However the KL loss enables us to use arbitrary architectural components inside the EBM network. We further add a section in appendix A.4 comparing the effect of backpropogating through all steps of Langevin compared to 1. We notice no quantitative impact when backpropogating through all steps of Langevin on MNIST compared to the last step, but note that full backpropogation is significantly more expensive.
>
> Q3) The multi-scale processing of EBMs has been explored in [1], which should be discussed and compared. Besides, [2][3][4] are relevant references of training EBMs that should be discussed.
>
> Thank you for this feedback. We agree the mentioned papers are related to our work and we have added and discussed them in the related work section.
>
> Q4) Qualitative speaking, long-run chains in figure 7 still have a trend of degradation from realistic images. Quantitative analysis (e.g., German-Rubin statistics) would be helpful for evaluating the long-run chains clearly.
>
> Thank you for this insightful feedback. We have added a plot showing the Inception scores of generated samples over time from an EBM trained with the KL+data augmentation and without using the KL+data augmentation in Figure 7. We find that an EBM trained with  KL+data augmentation exhibits a slower decay in Inception score of generations. An overall decay of Inception score over the number of steps of sampling is to be expected, as we are generating low temperature samples from our model, which thus have lower diversity. However, in comparison with our model without KL/data augmentation, our low temperature samples more consistently capture the underlying shape of objects. We unfortunately could not find a reference for German-Rubin statistics, but we are happy to include them if the reviewer could provide a reference.

---

### Official Review · AnonReviewer2 · 2020-10-31
**Interesting but the results are not strong**

**Rating:** 5
**Confidence:** 5

**Review:**

This paper proposes several techniques to improve contrastive divergence training of energy-based models (EBMs).
First, the paper proposes to estimate a gradient term, which is neglected in the standard contrastive divergence training method, and show that this correction avoids training instabilities in previous EBM training methods.
Other techniques include: using data augmentation, defining the energy function as a sum of energies over multi-scales, and using reservoir sampling.
Effects of each proposed techniques towards training EBMs are evaluated. The performance of the trained EBMs on image generation, OOD detection, and compositional generation are tested.

In generally, the paper is well written and addresses the important problem of improving EBM training. But I have some concerns.

1. It is not easy for general readers to understand the upper part of Figure 2, which is said to illustrate the overall effects of the losses L_CD and L_KL. What are the meanings of the red balls (dark and light) in the curve?

2. The paper overlooks a class of competitive training methods, which introduce auxiliary generators to train EBMs, including (Kim & Bengio, 2016; Kumar et al., 2019), [a] and so on.
The comment in Section 4 (related work), which describes these methods as utilizing pre-trained networks to approximate portions of energy training, is not correct (not capturing the core idea of these methods). Although learning EBMs without auxiliary generators is worthwhile exploring, the paper needs to give the readers an overall picture of the state-of-the-art of learning EBMs and does not give biased comments.

Although the proposed method is somewhat new, the results are not strong, which weakens the contribution of this paper. [a] achieved much better results than the proposed method in CIFAR-10 (Table 1). Additionally, computational cost of the proposed method should be given and compared to previous methods.
[a] Y. Song, Z. Ou. Learning Neural Random Fields with Inclusive Auxiliary Generators. arxiv 1806.00271, 2018.

SNGAN performs much better than the proposed method in CIFAR-10, but much worse in CelebA-HQ and LSUN. This may confuse readers. The "reimplementation of a SNGAN 128x128 model using the torch mimicry GAN library" in CelebA-HQ and LSUN may not faithfully reflect the performance of SNGAN.

3. Considering the above comment, the following claim in this paper needs revision.
"significantly outperforms past energy based approaches (with approximately the same number of parameters)"

4. The paper has sporadic writing problems.

Typo in Eq.(3)

divergenceterm

LSUN bedroom (?)

5. In A.4 (COMPARISON OF CD/KL GRADIENT MAGNITUDES), it is said that "the gradient of the KL objective is non-negligible". But it is this non-negligible gradient term that stabilize the EBM training. Need more analysis here.

How "Influences and relative magnitude of both loss terms" are calculated?

--------update after reading the response-----------

Thanks for the authors' response, but some non-trivial concerns are still not adequately addressed.
1) The inconsistent comparison results between SNGAN and the proposed method over CIFAR-10 and LSUN Bedroom datasets.
2) I can see the benefit such as compositionality from the proposed method of training EBMs. But the paper still seems to overlook the importance of giving the readers an overall picture of the state-of-the-art of learning EBMs. Table 1 should be expanded to include more state-of-the-art results from EBMs, whether using auxiliary generators or not.

---

> ### Author Response · Authors · 2020-11-20
> **Response**
>
> We thank the reviewer for their thorough and insightful review. We address major concerns below and have also correspondingly updated the text.
>
> Q1) It is not easy for general readers to understand the upper part of Figure 2, which is said to illustrate the overall effects of the losses L_CD and L_KL. What are the meanings of the red balls (dark and light) in the curve?
>
> We have redesigned Figure 2 (see updated paper). The intent of the red balls is to show that the L_KL encourages generated samples (dark red balls) to have low energy and high diversity (shown now by blue balls).
>
> Q2) The paper overlooks a class of competitive training methods, which introduce auxiliary generators to train EBMs, including (Kim & Bengio, 2016; Kumar et al., 2019), [a] and so on. The comment in Section 4 (related work), which describes these methods as utilizing pre-trained networks to approximate portions of energy training, is not correct (not capturing the core idea of these methods). Although learning EBMs without auxiliary generators is worthwhile exploring, the paper needs to give the readers an overall picture of the state-of-the-art of learning EBMs and does not give biased comments.
>
> We have revised our related work with comments about auxiliary generators. We agree the mentioned papers are related to our work and we have added them in the related work section.
>
> Q3) Although the proposed method is somewhat new, the results are not strong, which weakens the contribution of this paper. [a] achieved much better results than the proposed method in CIFAR-10 (Table 1). Additionally, computational cost of the proposed method should be given and compared to previous methods. [a] Y. Song, Z. Ou. Learning Neural Random Fields with Inclusive Auxiliary Generators. arxiv 1806.00271, 2018.
>
>
> We believe our approach gets the best generative performance out of approaches that do not use any auxiliary generator networks. Our implicit generation, compared to generation with auxiliary generators has unique benefits such as compositionality, which we show in section 3.3. Regarding performance, we have compared the effect of adding a KL loss in section 3.2. Our approach is roughly two times slower than past approaches based on implicit generation.
>
> How "Influences and relative magnitude of both loss terms" are calculated?
> We set the overall magnitude of both CD and KL loss terms to be 1 to 1, which the ratio indicated based off the derivation.
>
> We have revised our statement of “our approach significantly outperforming other past energy based approaches”, to “our approach significantly outperforms other implicit sampling based EBM approaches”. Such an approach has many desirable properties, such as compositionality and out of distribution robustness. We have also fixed typos mentioned.

---

### Author Response · Authors · 2020-11-24
**Re-evaluation Based on Rebuttal/Revision**

Dear Reviewers,

Thank you very much for your thorough and insightful review. We have revised our discussion of related work significantly and have also added clarifications and additional experiments on the effects of the KL loss towards performance. We believe our work shows very strong performance on interesting downstream tasks such as out-of-distribution detection and compositional generation (that we believe are the best that has been reported).

We spent a large amount of work answering the questions initially requested. We would appreciate it if you could take a look at the revised version and re-evaluate our work.

Thanks,
Paper Authors

---

### Decision · Program_Chairs · 2021-01-07
**Final Decision**

**Decision:**

Reject

**Comment:**

This paper introduces a bag of techniques to improve contrastive divergence training of energy-based models (EBMs), particularly a KL divergence term, data augmentation, multi-scale energy functions, and reservoir sampling. The overall paper is well written and clearly presented.

In response to the major concerns from reviewers, the AC recognizes the authors' effort in expanding related work and adding ablation on the effects of the KL loss. However, reviewers remain unconvinced by the significance of the current results. In particular, the quality improvement by adding the KL terms is subtle compared to using reservoir sampling (as evidenced in the contrast of the last two rows in Table 2). Moreover, the authors are also encouraged to compare additionally with recent development in EBM, as pointed out by R2 & R4.

The AC does find the results on downstream tasks such as out-of-distribution quite promising and interesting. Perhaps it's worth expanding the discussion with formal reasoning on why KL loss helps in this case.

All four knowledgeable reviewers are leaning towards rejection, the AC respects and agrees with the decision.